

# Classification of Alzheimer's disease stages from magnetic resonance images using deep learning

Alejandro Mora-Rubio[1], Mario Alejandro Bravo-Ortíz[1], Sebastián Quiñones Arredondo[1], Jose Manuel Saborit Torres[2], Gonzalo A. Ruz[3,4,5] and Reinel Tabares-Soto[1,5,6]

[1] Department of Electronics and Automation, Universidad Autonóma de Manizales, Manizales, Caldas, Colombia
[2] Unidad Mixta de Imagen Biomédica FISABIO-CIPF, Fundación para el Fomento de la Investigación Sanitario y Biomédica de la Comunidad Valenciana, Valencia, Spain
[3] Center of Applied Ecology and Sustainability (CAPES), Santiago, Chile
[4] Data Observatory Foundation, Santiago, Chile
[5] Facultad de Ingeniería y Ciencias, Universidad Asdolfo Ibáñez, Santiago, Chile
[6] Department of Systems and Informatics, Universidad de Caldas, Manizales, Caldas, Colombia

Corresponding author
Alejandro Mora-Rubio,
alejandro.morar@autonoma.edu.co

## ABSTRACT

Alzheimer's disease (AD) is a progressive type of dementia characterized by loss of memory and other cognitive abilities, including speech. Since AD is a progressive disease, detection in the early stages is essential for the appropriate care of the patient throughout its development, going from asymptomatic to a stage known as mild cognitive impairment (MCI), and then progressing to dementia and severe dementia; is worth mentioning that everyone suffers from cognitive impairment to some degree as we age, but the relevant task here is to identify which people are most likely to develop AD. Along with cognitive tests, evaluation of the brain morphology is the primary tool for AD diagnosis, where atrophy and loss of volume of the frontotemporal lobe are common features in patients who suffer from the disease. Regarding medical imaging techniques, magnetic resonance imaging (MRI) scans are one of the methods used by specialists to assess brain morphology. Recently, with the rise of deep learning (DL) and its successful implementation in medical imaging applications, it is of growing interest in the research community to develop computer-aided diagnosis systems that can help physicians to detect this disease, especially in the early stages where macroscopic changes are not so easily identified. This article presents a DL-based approach to classifying MRI scans in the different stages of AD, using a curated set of images from Alzheimer's Disease Neuroimaging Initiative and Open Access Series of Imaging Studies databases. Our methodology involves image pre-processing using FreeSurfer, spatial data-augmentation operations, such as rotation, flip, and random zoom during training, and state-of-the-art 3D convolutional neural networks such as EfficientNet, DenseNet, and a custom siamese network, as well as the relatively new approach of vision transformer architecture. With this approach, the best detection percentage among all four architectures was around 89% for AD *vs*. Control, 80% for Late MCI *vs*. Control, 66% for MCI *vs*. Control, and 67% for Early MCI *vs*. Control.

## INTRODUCTION

Alzheimer's disease (AD) is a progressive neurological disorder that primarily affects memory and cognition. Other symptoms, such as language disorders, hallucinations, and in some cases, seizures or parkinsonian features, may also be present (*Kumar et al., 2022*). One theory regarding the causes of AD links the $\beta$-amyloid peptide and mutations in its precursor proteins to weakened brain structures, which in turn impact various cognitive processes in the patient's brain. This theory is supported and assessed through two key markers: amyloid plaques and neurofibrillary tangles (*Ballard et al., 2011*).

Macroscopically, changes in brain morphology play a crucial role in the diagnosis of AD, with frontotemporal lobe atrophy being a common feature. This loss of brain volume in the region results in diminished function of the amygdala and hippocampus, which are responsible for memory processes. Medical imaging techniques, such as magnetic resonance imaging (MRI) and positron emission tomography (PET) scans, are utilized by specialists to evaluate the morphological changes in the brain caused by AD (*van Oostveen & de Lange, 2021*). In addition to brain scans, using cognitive tests, like the Mini-Mental State Examination (MMSE), that consider specific physical, psychological, and social skills is essential for a reliable diagnosis of the disease.

Regarding AD progression, as the disease advances the symptoms change, and a series of consecutive stages can be identified, following the convention used by Alzheimer's Disease Neuroimaging Initiative (ADNI, https://adni.loni.usc.edu). A brief description of each stage is presented below.

- Early mild cognitive impairment (EMCI): the affected person starts experiencing episodes of memory loss with words or the location of household items, nevertheless he can function independently and participate in social activities.
- Mild cognitive impairment (MCI): the affected people are susceptible to forgetting recent occurrences, becoming disoriented in their homes, and having difficulties with communication. This is often the longest stage of AD, lasting up to 4 years.
- Late mild cognitive impairment (LMCI): at this stage of the disease, patients may need help with daily tasks, facing increasing difficulty communicating and controlling their movements. Their memory and cognitive skills worsen, and changes in behavior and personality may occur.
- Alzheimer's disease (AD): as the disease progresses, the affected person requires increasing levels of attention and aid with daily tasks. This stage is characterized by growing unawareness of time and space, problems recognizing family and close friends, difficulty walking, and behavioral disturbances that may even lead to aggression.

Furthermore, with the rise of deep learning (DL) and its successful implementation in medical imaging applications, researchers have been working on automatic systems able to learn from MRI and PET scans to support the diagnosis of AD (*LaMontagne et al., 2019*; *Petersen et al., 2010*; *Zeng et al., 2018*). This kind of decision support system, created with artificial intelligence at its core, has been used successfully in different situations in the healthcare field improving diagnostic performance, reducing medical errors, and providing a better quality of service in developing countries (*Currie & Rohren, 2021*; *Porumb et al., 2020*). In general, DL techniques are developed to find patterns in the data and extract the relevant features for optimal classification (*Sharma, Sharma & Sharma, 2016*; *Chan et al., 2020*; *Bravo-Ortiz et al., 2021*), making them suitable for tasks with large volumes of labeled data based on human experiences, such as classifying the stages of AD using brain imaging (*Liu et al., 2014*; *Chitradevi & Prabha, 2020*; *Zeng, Li & Peng, 2021*). In particular, for AD diagnosis, most of the research papers consider a 3D convolutional neural network (CNN) approach to take advantage of the multidimensional features in brain scans (*Payan & Montana, 2015*; *Huang et al., 2019*). Alternatively, the work of *Cheng & Liu (2017)* proposes a 2D CNN approach, combined with recurrent neural networks to link the features of the three-dimensional scans. In the development of such systems, image pre-processing is key to achieving the best possible results, and in that regard, the main tool is FreeSurfer, which allows operations like skull suppression, bias field correction, anatomical registration, segmentation, reconstruction, and parcellation of the cortical surface (*Fischl, 2012*).

Acknowledging the growing number of patients with AD, and the importance of improving their quality of life, it is essential to develop more effective and efficient tools to diagnose the disease. This article presents a DL-based approach to the classification of MRI scans in the different stages of AD, using the EfficientNet, DenseNet, a custom Siamese architecture, and a Vision Transformer, the detection percentage among all four architectures was around 89% for AD *vs.* Control, 80% for Late MCI *vs.* Control, 66% for MCI *vs.* Control, and 67% for Early MCI *vs.* Control, and it is organized as follows: the Methods section describes the dataset, CNN architectures, and training set-up; the Results section presents the experiments and classification results obtained; the Discussion section reviews the results and their impact on the field; and finally the Conclusion of this work.

## METHODS

### Dataset

For this task, open-access imaging databases ADNI (https://adni.loni.usc.edu) and OASIS (*LaMontagne et al., 2019*) are the most commonly used sources of MRI scans to train and evaluate DL models. The ADNI was launched in 2003 as a public-private partnership, led by Principal Investigator Michael W. Weiner, MD. The primary goal of ADNI has been to test whether serial MRI, PET, other biological markers, and clinical and neuropsychological assessment can be combined to measure the progression of MCI and early AD.

For the purpose of this article, a dataset comprising images from both databases was used. There are a total of 2,559 images from 1,126 subjects, where the gender distribution is

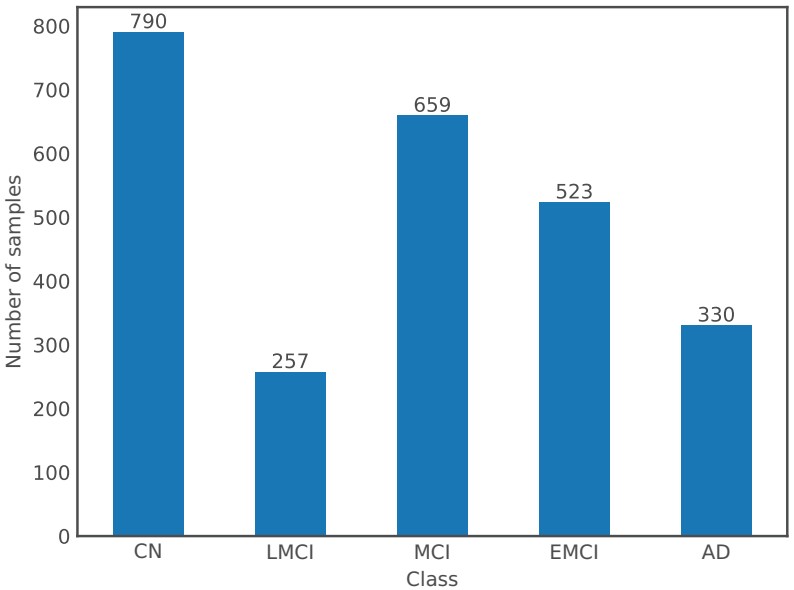

**Figure 1 Class distribution in the dataset.**

relatively even with 54% of images from males and the remaining 46% from females; and ages ranging from 55 to 96 years old with a mean of 75 years old. Following the labels used by ADNI, there are a total of five different classes in the dataset: Cognitively Normal or Control (CN), EMCI, MCI, LMCI, and AD; OASIS labels, which are expressed numerically according to the Clinical Dementia Rating, were changed to ADNI format: label 0 corresponds to CN, (0,1) to MCI, and [1,3] to AD. Figure 1 shows the number of samples per class in the dataset.

In order to reduce data-induced bias in the model, the images were partitioned into the training, validation, and test sets on a subject basis, obtaining a training set with 683 patients (1,569 images), a validation set with 222 patients (513 images), and a test set with 221 patients (477 images), where every session from a patient is contained in a single set. Additionally, a similar age distribution within each partition was also maintained (see Fig. 2).

## Deep learning architectures

### Siamese 3D

In this type of architecture, the data flows through parallel routes at the same time, which are then combined to make a final prediction. In this case, each path is composed of sets of 3D convolutional layers, 3D batch normalization, and average pooling operations, in the end, the feature maps are flattened, concatenated, and then passed to the fully connected layers for the final prediction. It is worth mentioning, that this is a custom architecture, designed for the classification of AD stages using 3D MRI scans (*Saborit Torres, 2019*). In the proposed configuration, this model has a total of 392,822 trainable parameters for the binary classification tasks, and 392,923 for the 3-way classification tasks. Figure 3 shows the model architecture.

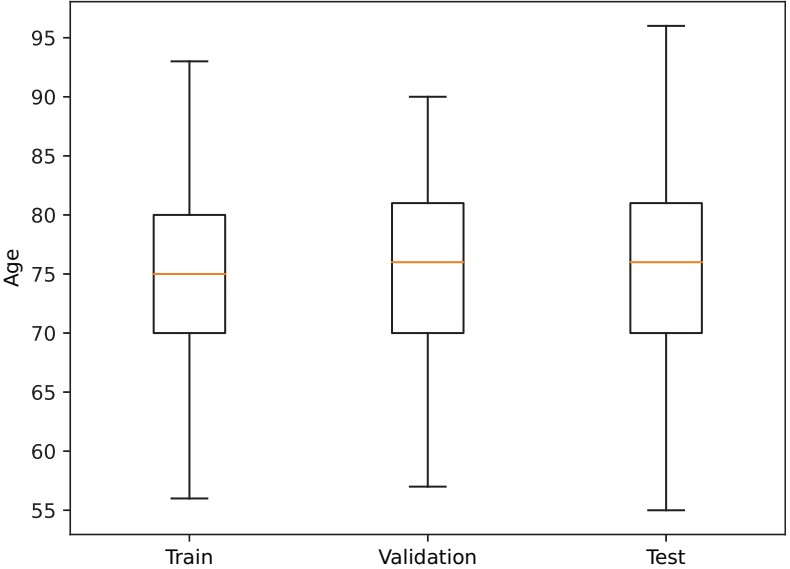

**Figure 2 Age distribution in each dataset partition.**

### Vision transformer

Vision transformers (ViT) (*Dosovitskiy et al., 2020*) are a groundbreaking approach to computer vision that has recently gained popularity due to their impressive performance on various tasks. Unlike traditional CNNs, which rely on hand-engineered feature maps, vision transformers use self-attention mechanisms to learn features directly from the input data dynamically. This approach has several advantages, including increased flexibility and scalability and improved performance on object detection, segmentation, and classification tasks. Additionally, ViT (*Khan et al., 2022*) has shown promise in domains beyond computer vision, including natural language processing and speech recognition. One of the critical innovations of ViT is self-attention, which allows the network to attend to different parts of the input data and dynamically adjust the importance of each feature. ViT (*Khan et al., 2022*) contrasts traditional CNNs, which rely on fixed, pre-defined feature maps that may only be optimal for some tasks. Another advantage of vision transformers is their ability to learn from large-scale datasets. Pre-training on massive datasets such as ImageNet has been shown to significantly improve the performance of ViT on downstream tasks, demonstrating the importance of unsupervised learning in deep learning models (*Dosovitskiy et al., 2020*). Despite their impressive performance, ViT is still a relatively new approach to computer vision, and there is ongoing research into how best to design and optimize these models. In the proposed configuration, this model has a total of 1,853,698 trainable parameters.

### DenseNet

DenseNet was originally proposed as a general-purpose image classification architecture, where the dense convolutional blocks improved significantly the vanishing gradient problem experienced by very deep neural networks. Each of these dense convolutional blocks refers to a set of convolutional layers that are all connected between them, this

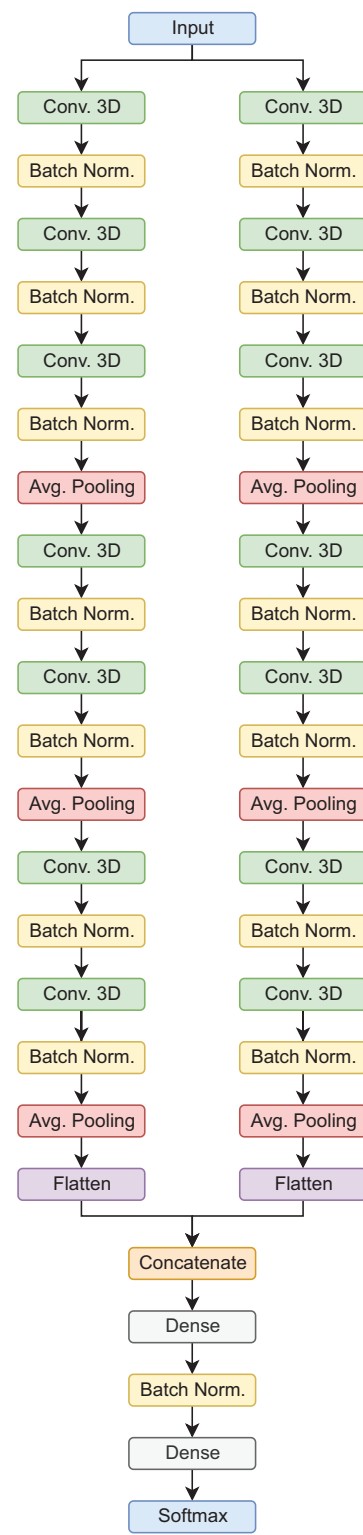

**Figure 3  Siamese 3D model architecture.**           

means that "the feature-maps of all preceding layers are used as inputs, and its own feature-maps are used as inputs into all subsequent layers" (*Huang et al., 2016*). By concatenating several of these blocks, the DenseNet architecture and all its variants are created. This network has been successfully used in different biomedical imaging classification problems, such as the detection of COVID-19 in computed tomography scans avoiding manual delineation of the lesions (*Xiao et al., 2022*), the detection of AD from brain scans based on features extracted from the hippocampal region (*Cui & Liu, 2019*), and the detection of skin cancer by classifying the images of skin lesions (*Villa-Pulgarin et al., 2022*), which motivate its inclusion in this work. In the proposed configuration, this model has a total of 11,244,674 trainable parameters for the binary classification tasks, and 11,245,699 for the 3-way classification tasks.

### EfficientNet

EfficientNet refers to a series of convolutional neural networks that scale the three dimensions of the network at the same time, achieving better classification results with fewer trainable parameters compared to other state-of-the-art architectures. The dimensions affected are depth (number of layers), width (number of channels or filters per layer), and resolution (size of the input images and feature maps). In particular, the B7 variant was used for this article, which is the biggest one as presented in the original paper (*Tan & Le, 2019*). As for DenseNet, this architecture has been implemented in various tasks of biomedical image processing such as the detection of diabetic retinopathy from digital fundus images (*Jaiswal et al., 2021*; *Wang et al., 2020*), the detection of tuberculosis in chest X-ray images (*Munadi et al., 2020*), and aging classification based on skin microstructure characteristics (*Moon & Lee, 2022*). In the proposed configuration, this model has a total of 68,764,114 trainable parameters for the binary classification tasks, and 68,766,675 for the 3-way classification tasks.

## Training procedure

Regarding the model training set-up, the MONAI (*MONAI Consortium, 2022*) was used, it is based on PyTorch and allows easy implementation of input data pipelines with image intensity scaling to the [0,1] range, image resizing to $91 \times 91 \times 91$ pixels, and spatial data-augmentation operations, such as random 90° rotation, random flip, and random zoom. The latter operations aim to improve the models' generalization ability by adding variability to the images during training. In terms of training hyperparameters, all models were trained for 250 epochs, using a weighted cross entropy loss to account for data imbalance, and the Adadelta optimizer with a learning rate of 1.0, $\rho = 0.95$, and $\varepsilon = 1 \times 10^{-7}$. All the experiments were performed using NVIDIA Quadro RTX 8000 graphic processor units.

## RESULTS

This section presents the experiments performed over the previously described dataset, and the classification results achieved for each one of them. The first experiment consists of the binary classification between the Control cases and AD, which is the benchmark task in

**Table 1 Classification results for two classes—control and Alzheimer's disease.** The best results for each test metric is indicated in bold italics.

| Model | Train accuracy | Validation accuracy | Test accuracy | Test sensitivity | Test specificity |
|---|---|---|---|---|---|
| Siamese 3D | 0.9844 | 0.8646 | 0.7448 | 0.6776 | 0.5943 |
| ViT | 0.7804 | 0.9203 | *0.8902* | *0.8902* | *0.7401* |
| DenseNet | 0.9656 | 0.8854 | 0.7448 | 0.6322 | 0.5905 |
| EfficientNet B7 | 0.9672 | 0.8750 | 0.8542 | 0.8085 | 0.6835 |

AD detection. Table 1 presents the training, validation, and test results achieved for this task, where the ViT architecture performed the best.

Figure 4A shows the receiver operator curve for the CN *vs*. AD task using the predictions from the EfficientNet model.

The second experiment involves the binary classification between Control cases against LMCI. Table 2 presents the results for this experiment.

Figure 4B shows the receiver operator curve for the CN *vs*. LMCI task using the predictions from the DenseNet model.

Similarly, the third experiment consists of the classification of Control cases against MCI. Table 3 presents the results for this experiment.

Figure 4C shows the receiver operator curve for the CN *vs*. MCI task using the predictions from the DenseNet model.

The fourth experiment consists of the classification of Control and EMCI, which is significantly more challenging since the physical effect of the disease is not as aggressive as for AD, but ultimately it is more relevant for physicians since it would help them identify potential AD patients in the early stages. Table 4 presents the results for this experiment.

Figure 4D shows the receiver operator curve for the CN *vs*. EMCI task using the predictions from the Siamese 3D model.

The fifth experiment aimed to classify the early and late stages of the disease progression; the early stages were the Control and EMCI cases, and MCI, LMCI, and AD were considered as late stages. The main goal was to take advantage of the complete dataset aiming to improve classification results. Table 5 presents the results for the fourth experiment.

The sixth and final experiment consists of a multi-class classification problem involving Control, MCI (including early and late MCI), and AD cases. Table 6 presents the results of this experiment.

## DISCUSSION

This article presents a DL-based approach to the classification of MRI scans in the different stages of AD, using a curated set of images from both ADNI and OASIS databases. There are studies from different countries around the world that suggest an increasing prevalence of dementia in older patients, and how AD can be considered one of the leading causes of it (*Rasmussen & Langerman, 2019*), it is important for the scientific and healthcare community to develop and improve current diagnostic techniques, in particular during its

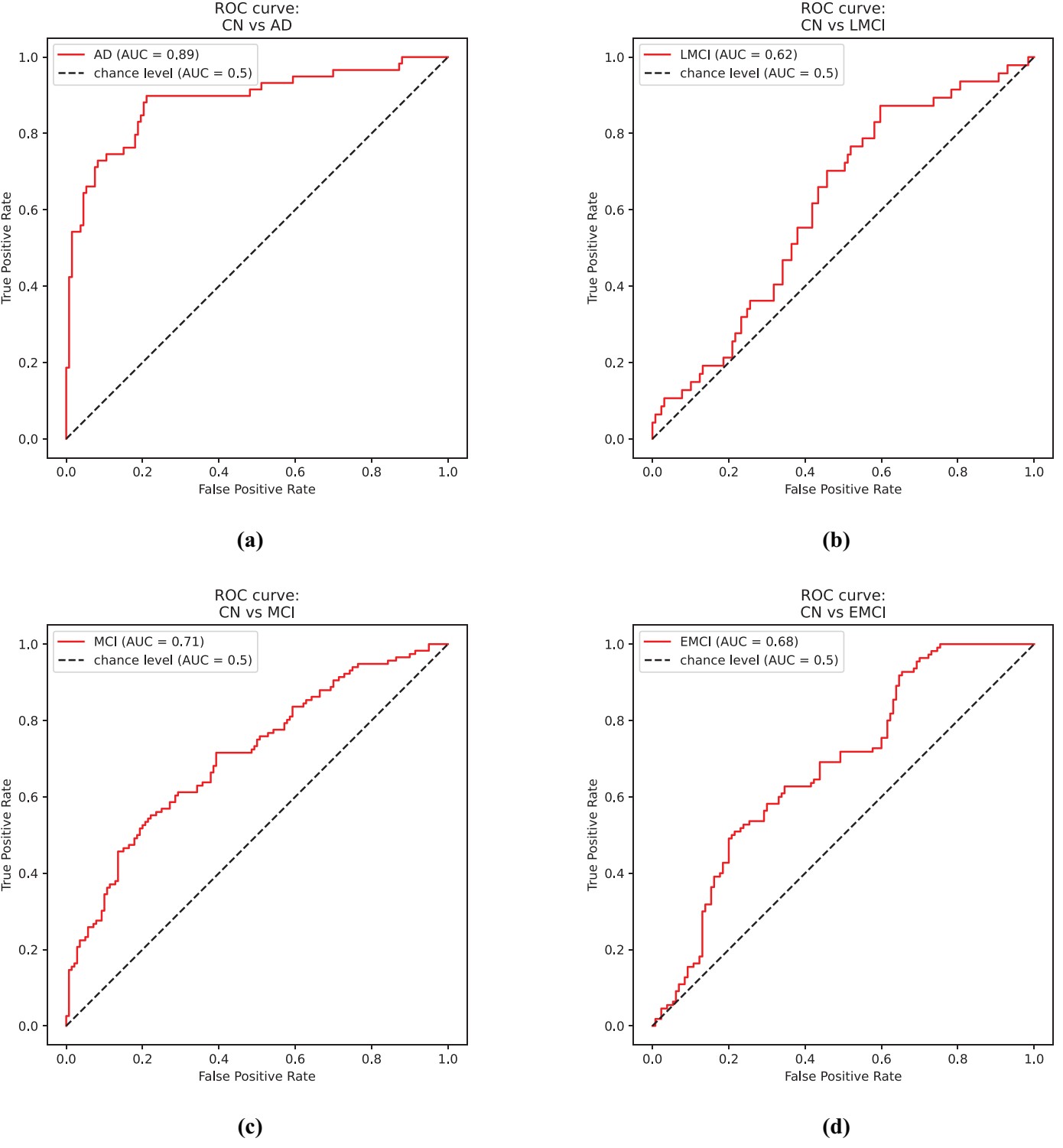

**Figure 4 ROC curves.** (A) Control and Alzheimer's disease. (B) Control and late mild cognitive impairment. (C) Control and mild cognitive impairment. (D) Control and early mild cognitive impairment. 

**Table 2 Classification results for two classes—control and late mild cognitive impairment.** The best results for each test metric is indicated in bold italics.

| Model | Train accuracy | Validation accuracy | Test accuracy | Test sensitivity | Test specificity |
|---|---|---|---|---|---|
| Siamese 3D | 0.7656 | 0.8021 | 0.7578 | 0.5156 | 0.5156 |
| ViT | 0.7656 | 0.9070 | *0.8056* | 0.6870 | 0.6530 |
| DenseNet | 0.9984 | 0.8385 | 0.7031 | *0.6991* | *0.6633* |
| EfficientNet B7 | 0.7375 | 0.8177 | 0.7734 | 0.5833 | 0.5833 |

**Table 3 Classification results for two classes—control and mild cognitive impairment.** The best results for each test metric is indicated in bold italics.

| Model | Train accuracy | Validation accuracy | Test accuracy | Test sensitivity | Test specificity |
|---|---|---|---|---|---|
| Siamese 3D | 0.8966 | 0.7305 | 0.6250 | 0.6017 | 0.6017 |
| ViT | 0.8506 | 0.7103 | 0.6432 | 0.5970 | 0.6420 |
| DenseNet | 0.9976 | 0.7188 | *0.6641* | 0.6449 | *0.6633* |
| EfficientNet B7 | 0.9976 | 0.7383 | *0.6641* | *0.6530* | 0.6591 |

**Table 4 Classification results for two classes—control and early mild cognitive impairment.** The best results for each test metric is indicated in bold italics.

| Model | Train accuracy | Validation accuracy | Test accuracy | Test sensitivity | Test specificity |
|---|---|---|---|---|---|
| Siamese 3D | 0.7331 | 0.7344 | *0.6719* | *0.6749* | *0.6511* |
| ViT | 0.7932 | 0.7020 | 0.6320 | 0.6720 | 0.6311 |
| DenseNet | 0.9596 | 0.7383 | 0.5938 | 0.6094 | 0.5913 |
| EfficientNet B7 | 0.5807 | 0.6523 | 0.5042 | 0.4915 | 0.4640 |

**Table 5 Classification results for two classes—early and late stages.** The best results for each test metric is indicated in bold italics.

| Model | Train accuracy | Validation accuracy | Test accuracy | Test sensitivity | Test specificity |
|---|---|---|---|---|---|
| Siamese 3D | 0.8542 | 0.6875 | 0.6473 | 0.6292 | 0.6292 |
| ViT | 0.7920 | 0.6990 | 0.6540 | 0.6560 | 0.6520 |
| DenseNet | 1.0000 | 0.7051 | 0.6540 | 0.6505 | 0.6505 |
| EfficientNet B7 | 0.7760 | 0.7246 | *0.6652* | *0.6550* | *0.6550* |

**Table 6 Classification results for three classes—control, mild cognitive impairment, and Alzheimer's disease.** The best results for each test metric is indicated in bold italics.

| Model | Train accuracy | Validation accuracy | Test accuracy | Test sensitivity | Test specificity |
|---|---|---|---|---|---|
| Siamese 3D | 0.6250 | 0.5645 | 0.5625 | 0.3585 | 0.6664 |
| ViT | 0.6120 | 0.5780 | 0.5510 | *0.4260* | 0.6650 |
| DenseNet | 0.5964 | 0.5820 | 0.5446 | 0.4199 | *0.6861* |
| EfficientNet B7 | 0.5443 | 0.5703 | *0.5893* | 0.4163 | 0.6765 |

early stages. Since there is yet to be a treatment that can revert the physiological effects of severe dementia, the early diagnosis of the disease is imperative for appropriate care of the patient, and a realistic projection of the following years and stages. In this regard, brain imaging and CAD systems are of high interest to researchers, where the high capacity of DL algorithms to extract relevant features from images can be leveraged to detect the subtle changes present in the early stages of the disease.

Our methodology encompasses image pre-processing using FreeSurfer, spatial data augmentation techniques such as rotation, flip, and random zoom during training, and state-of-the-art 3D CNNs, including EfficientNet, DenseNet, and a custom siamese network, as well as the relatively new approach of Vision Transformer architecture. Using this approach, the best detection percentages among all three architectures were approximately 89% for AD *vs.* Control, 80% for Late MCI *vs.* Control, 66% for MCI *vs.* Control, and 67% for Early MCI *vs.* Control. It is noteworthy that all three models involved in the experiments achieve competitive results and outperform one another in specific cases, with the Siamese 3D model's performance on the CN *vs.* Early MCI task and the ViT model's performance on the CN *vs.* AD task being particularly remarkable. Regarding these results, it is essential to acknowledge the difficulty in comparing them with similar research papers due to the numerous factors affecting model performance. If these factors are not adequately reported, reproducing the results and directly comparing them with ours becomes nearly impossible. Consequently, the dataset and selected partition are crucial for obtaining reliable results that accurately represent how the model would perform in real-life scenarios. As reported in medical tasks, a significant portion of predictive models presented in the scientific literature offer little to no value to the healthcare community due to the presence of bias in the models, which affects the accuracy of predictions (*Roberts et al., 2021*, *Wynants et al., 2020*). Bearing this in mind, the code and references to the data used during the experimentation stage of our work are available in a GitHub repository (https://github.com/MoraRubio/alzheimer-stages-dl), and this article describes, to the best of our ability, the materials and methods used throughout the process.

In particular, compared to the referenced works on classification using 3D CNNs, The work by *Huang et al. (2019)* also includes FDG-PET images, which offer relevant information for the diagnosis, but at a greater cost for the patient or healthcare system, resulting in less viable option for developing countries with lower resources, as well as, fewer images for model training and evaluation. On the other hand, the work by *Payan & Montana (2015)*, uses an autoencoder for feature extraction instead of feeding the images directly to the CNN, which adds extra complexity for model development and deployment. In this work, only MRI images are considered, since they are more common and accessible than PET images, and the images are directly fed to the CNN architectures. As mentioned above, a direct comparison of the detection percentages is not easily performed.

Acknowledging the limited size of the involved dataset, and the number of DL methods; future work should be aimed at exploring different network architectures, such as Convolutional Vision Transformers, to enhance the detection accuracy of the different stages. Initially, this will focus on binary classification to differentiate intermediate stages

from Control and AD patients. However, the ultimate goal is to identify those MCI patients who are most likely to develop AD, particularly in the early stages, where prediction is crucial for healthcare professionals in providing optimal patient care. Moreover, assembling a larger image dataset while adhering to ethical principles (*Tabares-Soto et al., 2022*) and aiming to minimize data-induced bias in the models can positively impact the detection capabilities and, more importantly, the generalization ability of the models, which is essential for developing viable CAD systems. Furthermore, it is vital to conduct in-depth analysis of the results using visualization or interpretation techniques, such as activation maps, occlusion sensitivity, or gradient-based heat maps. These methods could enable comparison with physical markers assessed by medical specialists or even identify new regions of interest for AD detection.

## CONCLUSIONS

Early diagnosis of AD is key to providing adequate care to the patient, but existing detection techniques are not definitive enough to provide a certain diagnosis. The evaluation of brain morphology and its changes through time is one of the tools used by specialists to diagnose the disease, however, changes in the early stages are difficult to identify in plain sight, and the use of CAD systems is of great interest for this task. The results presented in this article regarding the classification of different stages of AD using brain MRI indicate that DL models do have the ability to correctly identify control/healthy patients from the ones suffering from dementia, nevertheless, there is still a lot of work to be done in order to create reliable models and test them appropriately in real-life scenarios.

## ACKNOWLEDGEMENTS

Data were provided in part by OASIS-3: Longitudinal Multimodal Neuroimaging: Principal Investigators: T. Benzinger, D. Marcus, J. Morris; NIH P30 AG066444, P50 AG00561, P30 NS09857781, P01 AG026276, P01 AG003991, R01 AG043434, UL1 TR000448, R01 EB009352. AV-45 doses were provided by Avid Radiopharmaceuticals, a wholly owned subsidiary of Eli Lilly. ADNI data are disseminated by the Laboratory for Neuro Imaging at the University of Southern California.

### Funding

This work was supported by the Universidad Autonoma de Manizales as part of the project "Detección de COVID-19 en imágenes de rayos X usando redes neuronales convolucionales" with code 699-106, and also to the projects "CH-T1246: Oportunidades de Mercado para las Empresas de Tecnología—Compras Públicas de Algoritmos Responsables, Éticos y Transparentes", ANID PIA/BASAL FB0002, and ANID/PIA/ANILLOS ACT210096. Data collection and sharing for this project was funded by the Alzheimer's Disease Neuroimaging Initiative (ADNI) (National Institutes of Health Grant U01 AG024904) and DOD ADNI (Department of Defense award number W81XWH-12-2-0012). ADNI is funded by the National Institute on Aging, the National Institute of

Biomedical Imaging and Bioengineering, and through generous contributions from the following: AbbVie, Alzheimer's Association; Alzheimer's Drug Discovery Foundation; Araclon Biotech; BioClinica, Inc.; Biogen; Bristol-Myers Squibb Company; CereSpir, Inc.; Cogstate; Eisai Inc.; Elan Pharmaceuticals, Inc.; Eli Lilly and Company; EuroImmun; F. Hoffmann-La Roche Ltd and its affiliated company Genentech, Inc.; Fujirebio; GE Healthcare; IXICO Ltd.; Janssen Alzheimer Immunotherapy Research & Development, LLC.; Johnson & Johnson Pharmaceutical Research & Development LLC.; Lumosity; Lundbeck; Merck & Co., Inc.; Meso Scale Diagnostics, LLC.; NeuroRx Research; Neurotrack Technologies; Novartis Pharmaceuticals Corporation; Pfizer Inc.; Piramal Imaging; Servier; Takeda Pharmaceutical Company; and Transition Therapeutics. The Canadian Institutes of Health Research is providing funds to support ADNI clinical sites in Canada. Private sector contributions are facilitated by the Foundation for the National Institutes of Health (www.fnih.org). The grantee organization is the Northern California Institute for Research and Education, and the study is coordinated by the Alzheimer's Therapeutic Research Institute at the University of Southern California. There was no additional external funding received for this study. The funders had no role in study design, data collection and analysis, decision to publish, or preparation of the manuscript.

## Grant Disclosures

The following grant information was disclosed by the authors:
Universidad Autonoma de Manizales: 699-106, ANID PIA/BASAL FB0002 and ANID/PIA/ANILLOS ACT210096.
National Institutes of Health Grant: U01 AG024904.
DOD ANID PIA/BASAL FB0002.
ANID/PIA/ANILLOS ACT210096.
National Institutes of Health Grant: U01 AG024904.
DOD ADNI (Department of Defense): W81XWH-12-2-0012.
National Institute on Aging.
National Institute of Biomedical Imaging and Bioengineering.
AbbVie, Alzheimer's Association.
Alzheimer's Drug Discovery Foundation.
Araclon Biotech.
BioClinica, Inc.
Biogen.
Bristol-Myers Squibb Company.
CereSpir, Inc.
Cogstate.
Eisai Inc.
Elan Pharmaceuticals, Inc.
Eli Lilly and Company.
EuroImmun.
F. Hoffmann-La Roche Ltd.
Genentech, Inc.

Fujirebio.
GE Healthcare.
IXICO Ltd.
Janssen Alzheimer Immunotherapy Research & Development, LLC.
Johnson & Johnson Pharmaceutical Research & Development LLC.
Lumosity.
Lundbeck.
Merck & Co., Inc.
Meso Scale Diagnostics, LLC.
NeuroRx Research.
Neurotrack Technologies.
Novartis Pharmaceuticals Corporation.
Pfizer Inc.
Piramal Imaging.
Servier.
Takeda Pharmaceutical Company.
Transition Therapeutics.
The Canadian Institutes of Health Research.
Foundation for the National Institutes of Health.
Northern California Institute for Research and Education.
Alzheimer's Therapeutic Research Institute at the University of Southern California.
University of Southern California.
Avid Radiopharmaceuticals.

## Competing Interests

The authors declare that they have no competing interests.

## Author Contributions

- Alejandro Mora-Rubio conceived and designed the experiments, performed the experiments, analyzed the data, performed the computation work, prepared figures and/or tables, authored or reviewed drafts of the article, and approved the final draft.
- Mario Alejandro Bravo-Ortíz performed the experiments, analyzed the data, performed the computation work, prepared figures and/or tables, and approved the final draft.
- Sebastián Quiñones Arredondo performed the experiments, analyzed the data, performed the computation work, prepared figures and/or tables, and approved the final draft.
- Jose Manuel Saborit Torres analyzed the data, prepared figures and/or tables, data collection and preparation, and approved the final draft.
- Gonzalo A. Ruz conceived and designed the experiments, authored or reviewed drafts of the article, and approved the final draft.
- Reinel Tabares-Soto conceived and designed the experiments, authored or reviewed drafts of the article, and approved the final draft.

## Data Availability

The data is available from ADNI (https://adni.loni.usc.edu) and OASIS (https://www.oasis-brains.org).

The search parameters for ADNI database are available at Zenodo: Alejandro Mora-Rubio. (2023). MoraRubio/alzheimer-stages-dl: First Release (v1.0). Zenodo. https://doi.org/10.5281/zenodo.7855386.

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
