# Peer review of "Classification of Alzheimer’s disease stages from magnetic resonance images using deep learning"

_PeerJ Computer Science, doi:10.7717/peerj-cs.1490_

## Round 0.1 · original submission · Major Revisions

The manuscript received criticism from 3 reviewers. The reviewers have raised some concerns about the current version and asked for some amendments. I invite the authors to carefully address the reviewers' concerns (particularly, Reviewer 1) before resubmission.

·

Basic reporting

Lines 82-83
The manuscript could be improved by explicitly describing the differences between the proposed work and those of Payan and Montana (2015), and Huang et al. (2019), and how this work approached/solved their potential deficiencies.

Lines 38-39, 91-92, 200-201
Personally, I consider the way the numbers are reported here a bit misleading, as they refer to the best result for each case, and not the overall result of the best architecture.
My point is: how should one use these different architectures in practice? I don't instantly see an optimal solution for choosing the right network in each case, as probably one would choose a single architecture to be used in practice. I recommend the authors to either pick one network as the best, or clearly state that these results correspond to the best option among all three.

Lines 112-113
In my opinion, Figure 1 would be better visualized if Classes were ordered by disease stages, rather than number of samples.

Line 119
In computer vision, "siamese network" is a rather established term and people should be able to understand the meaning and implications without any problem. However, the term "bilinear" might cause some confusion as it's usually associated with "bilinear filters/mapping", which, as I understand it, is not what is meant here. Please consider renaming.

Lines 211-212
The GitHub repository link should be added as a footnote or in parenthesis.

Line 214
Specifically about Vision Transformers, a quick search on Google Scholar with the terms "vision transformer alzheimer" shows 9000+ results. Please consider including one of them, or maybe changing the example.

Lines 56-57, 98
Besides linking the ADNI's website, please also cite one of their recent papers, for proper crediting of the authors.

Lines 4-6, 97-
Please make sure that the EULA of each dataset is being followed. For instance, I believe it is necessary to cite ADNI right after the authors, and to include a given text in the Datasets section.

Lines 76-77, 80
Parentheses are misplaced in some of the references.

Experimental design

The authors experiment with some CNN architecture for the AD classification task.
This is an interesting topic; however, the research question was not clearly stated. This also applies to how this paper fills the knowledge gap in the literature. Please make sure to clearly state the research question, as well as how this paper fills the knowledge gap in the literature.

Lines 86-87
FreeSurfer is mentioned a number of times, but details are never provided.
What were the applied operations? What were the (hyper-)parameters? Is everything default or is anything adjusted?
Providing the code to run the FreeSurfer pipeline/stage in the GitHub repository is also recommended.

Line 107
LaMontagne et al. (2019) says that ages ranged from 42 to 95 years.
Given this, how were the datasets used/filtered? This question specifically comes from "age", but could spread to other aspects. Please clarify how datasets were filtered, including all considered options and parameters.

Lines 151-152
This is just a provocative commentary/question for the authors. I understand that "spatial data-augmentation operations'' are the standard in general computer vision. However, I imagine how relevant they are in this scenario where every image is standardized. For comparison, how useful would it be to train a car detector with augmented upside-down images? There should be papers about this.

Validity of the findings

There are two major additions that could greatly improve the paper quality.
One of them is adopting one of the many visualization and explanation techniques for CNNs. Some of them are also readily available in MONAI. For instance, which areas of the brain are the ones considered most important for the classification? Are they related to the ones considered by medical experts?
The other one is already stated by the authors in lines 203-206, which is the general lack of comparison in this field. This could be solved if researchers started adopting standardized and open datasets/frameworks, such as CADDementia (https://caddementia.grand-challenge.org/) and TADPOLE (https://tadpole.grand-challenge.org/).
As one additional note, I would personally give more emphasis to the 3-way classification as it more closely relates to the real practice/application.

Additional comments

In my opinion, the paper has its merits, and would get even better if authors took into account the suggestions made here.

On a side note, this is a comment that authors might consider in future works. CNNs with a VGG-style architecture tend to have around 80% of the total number of their parameters in the first fully connected layer, depending on the number of hidden units, and the dimensions of the previous layer. In later architectures, such as GoogLeNet and ResNet, this was solved by replacing this fully connected layer with an average pooling layer. I suggest the authors evaluate this number in their custom network, and consider this suggestion in the future.

Reviewer 2 ·

Basic reporting

please see the overall comments in section 4 Additional Comments.

Experimental design

please see the overall comments in section 4 Additional Comments.

Validity of the findings

please see the overall comments in section 4 Additional Comments.

Additional comments

The paper suggests the use of deep learning models like DenseNet, EfficientNet, and a siamese network for the classification of the various stages of Alzheimer's disease. The paper addresses a problem that is very appropriate in the care and diagnosis of patients suffering from AD by identifying various stages of AD. The paper presents the methodology as preprocessing techniques followed by various data-augmentation operations. While the paper discusses the problem that has merits, there are some major drawbacks to the paper.
1. The use of efficientNet, DenseNet, and Siamese network isn't new in the identification of the various stages of AD.
2. The paper talks about the methodology in its abstract section. The authors have mentioned various data-augmentation operations. The operations aren't mentioned well in the paper.
3. The paper is not well written. The introduction seems to be poorly written and needs rewriting.
4. The technical merits of the paper are very weak considering that the preprocessing techniques aren't well mentioned and the algorithms are just off-the-shelf algorithms found publicly. There is no novelty in terms of data augmentation and algorithmic implementation.

Reviewer 3 ·

Basic reporting

no comment

Experimental design

no comment

Validity of the findings

no comment

Additional comments

This paper is nice and has mathematical real-world applications; I see many positive aspects in this work and would like to see it published. In fact, this paper will be of value and interest to as a significant portion of potential readers of the journal. In my opinion, this paper can be further improved in the following aspects:

1. The contributions should be more clearly explained with more details on how to improve the existing results, especially in the references the authors cited.

2. Some parts of mathematic derivations are not given in details. I suggest the authors a carefully checking and give all the necessary manipulations for the derived formulas.

3. More detailed review of the literature is expected in separate section. Specially, it is required that the previous solutions to this problem be addressed. Then, the advantages (and disadvantages?) of the proposed methods and should be discussed.

4. Some latest references about AD and AI should be added to give readers an up-to-date picture. In this sense, the following papers can be referred: [a1] A new deep belief network-based multi-task learning for diagnosis of Alzheimer’s disease, Neural Computing and Applications; [a2] A new switching-delayed -PSO-based optimized SVM algorithm for diagnosis of Alzheimer's disease, Neurocomputing; [a3] AGGN: Attention-based Glioma Grading Network with Multi-scale Feature Extraction and Multi-modal Information Fusion, Computers in Biology and Medicine

5. The authors should clarify if the proposed method requires data processing or not. More detail information should be added to make clear explanations.

6. Some future directions can be discussed in the conclusion part.

7. The authors still need a careful check of English, formulas and format/style.

I recommend the publication of this paper after the authors address the above concerns.

---

## Round 0.2 · Minor Revisions

I think the manuscript has been satisfactorily revised. However, the authors have a few more comments to address in their final version. Below are my comments.

1. Including the number of parameters for the deep models in the comparison or separate table will be interesting. And, preferably, other computational costs.

2. Combining Figures 4-7 in one figure seems better to me and the readers. It is even more professional.

3. Please mention the limitations of the current work and the potential solutions for future work.

4. No need to redefine previous abbreviations. For instance, MRI and PET have been defined several times.

Reviewer 2 ·

Basic reporting

I would like to commend the authors for their prompt response and proactive efforts in addressing the feedback and suggestions provided during the review process. They have made significant improvements to the manuscript, including incorporating recent articles related to the research topic and rectifying grammatical mistakes and typos, resulting in a strengthened paper. The final version of this paper will be a valuable addition to the journal.

Experimental design

Good

Validity of the findings

good

Additional comments

please refer to the above comments.

---

## Round 0.3 · accepted · Accept

The manuscript has been revised and reviewers' and editor's concerns have been addressed in the revised version.